# Blast Resistance of Reinforced Concrete Slabs Based on Residual Load-Bearing Capacity

**DOI:** 10.3390/ma15186449

**Published:** 2022-09-16

**Authors:** Lijun Wang, Shuai Cheng, Zhen Liao, Wenjun Yin, Kai Liu, Long Ma, Tao Wang, Dezhi Zhang

**Affiliations:** 1School of Nuclear Engineering, Rocket Force University of Engineering, Xi’an 710024, China; 2Northwest Institute of Nuclear Technology, Xi’an 710024, China

**Keywords:** reinforced concrete slab, blast loading, residual load-bearing capacity, dynamic response, blast resistance

## Abstract

In this paper, the blast-loading experiment and numerical simulation are carried out for RC slabs with two typical reinforcement ratios. The time history of reflected shockwave pressures and displacement responses at different positions on the impact surface of the specimens are obtained, and the influence of the reinforcement ratio on the dynamic responses and failure modes of the RC slabs is analyzed. Based on the experimental data, the simulation model of the RC slab is verified, and the results indicate good agreement between the two methods. On this basis, the residual load-bearing capacity of the damaged RC slabs is analyzed. The results show that the load distribution on the impact surface of the slab is extremely uneven under close-in blast loading. The resistance curve shape of the RC slabs varies markedly before and after blast loading, and its load bearing capacity and bending stiffness deteriorate irreversibly. Increasing the reinforcement ratio can impede crack extension, reduce the slab’s residual displacement, and, at the same time, reduce the decrease of the damaged slab’s load-bearing capacity. The findings of this study will provide insights into the anti-explosion design and damage evaluation of RC slabs.

## 1. Introduction

The reinforced concrete (RC) slab is a typical load-bearing structural member in reinforced concrete buildings. Due to the influence of factors such as structural layer thickness, reinforcement ratio, and impact surface area, RC slabs tend to undergo larger deformation, more severe failure, and other forms of damage than RC beams/columns under the blast shockwave loading of the same intensity [1,2,3]. Figuring out the dynamic response and damage characteristics of RC slabs under shockwave loading induced by chemical explosions, and quantitatively and accurately evaluating the degradation of their load-bearing capacity, are of referential significance for the anti-explosion design of RC slabs and the evaluation of damage effects of weapons.

At present, considerable research efforts have been made by researchers at home and abroad concerning the failure modes and dynamic response of RC slabs under blast loading, yielding a series of results [4,5,6]. Lan et al. [7] conducted an experimental investigation of 74 groups of RC slabs under different charge quantities and stand-off distances and analyzed the differences of their damage and failure modes. Huff [8] also systematically studied the failure modes of a typical two-way RC slab under blast loading by experiments. Wang Wei et al. [9,10] carried out a series of experiments and numerical simulations to study the damage and failure of square RC slabs subjected to close-in explosions and obtained their damage–failure modes and criteria. Most experimental studies have also focused on determining new technologies to improve the deflection response and damage resistance of reinforced concrete slabs under explosive loads [11,12,13,14]. With respect to damage assessment of RC slabs, the P-I curve is often employed to assess the damage of structural members under blast loading [15,16,17,18]. As the equivalent single-degree-of-freedom (SDOF) analysis is simple and practical, it has been widely used in component damage assessment [17,19,20,21].

In fact, the dynamic response and failure modes of RC slabs are very complex because of the different blast-loading conditions (charge type, charge shape, scaled distance, etc.) and properties of the structure itself (concrete strength, reinforcement ratio, stirrup ratio, support boundary, etc.) [22,23,24]. Currently, the anti-explosion performance of RC slabs is evaluated mainly based on their dynamic response and the macro-damage modes of the damaged members. However, for the uneven load distribution in RC slabs subjected to close-in explosions, few analyses has been done using the performance index of the residual load-bearing capacity of damaged RC slabs to quantitatively evaluate the degradation of their bearing capacity.

In this paper, blast-loading experiments are conducted for RC slabs with two typical reinforcement ratios. The time history of the reflected shockwave pressures and displacement responses at different positions on the impact surface of the slabs are obtained, and the differences of the macro-damage modes of the two RC slabs at the same scaled distance are analyzed. The numerical simulation model of the slabs is constructed with finite element program LS-DYNA and corrected by the experimental data. On this basis, the residual load-bearing capacity of the damaged RC slabs with different reinforcement ratios is analyzed, the quantitative residual load-bearing capacity results are given, and the damage degrees of slab members with different reinforcement ratios are compared and analyzed.

## 2. Experimental Study

### 2.1. Test Specimen Design

According to the code for the design of concrete structures in China (2011 edition), RC slabs with two typical reinforcement ratios are designed and their geometric dimensions are both 1200 × 500 × 100 mm (length × width × thickness, respectively). The reinforcement drawings of the test specimens are shown in Figure 1 and both use double-layer two-way reinforcement. When the slab thickness is 100 mm~150 mm, the common diameter of the stressed reinforcement in the slab is 8~12 mm. Therefore, the longitudinal reinforcement bar diameters of the two RC slabs are 12 mm and 8 mm, and the corresponding reinforcement ratios are 1.41% and 0.63%. The stirrup diameter is 4 mm. The longitudinal bar is an HRB400 steel bar, the stirrup is an HPB235 steel bar, the concrete grade is C30, and the thickness of the protective layer is 50 mm.

During the experiment, the two RC slabs are placed vertically through independently designed supporting tools, as shown in Figure 2.

Slab A with the low reinforcement ratio, slab B with the high reinforcement ratio, and the pressure test tool are arranged in an equilateral triangle around the charge, with an interval of about 120° between them. The tool is fixed on the ground by multiple pegs and reinforced by welding the angle iron to the structure to ensure that the supporting tool does not move during the experiment. The four corners of the slab specimens are all clamped to give them full restraints. Steel plates are welded between the clamped corners to strengthen the restraint effect, so the boundary conditions can be regarded as fixed supports on four sides. The explosive used is a bare spherical TNT-RDX-Al explosive charge and its way of initiation is central detonation. The mass ratio of TNT to RDX was 4:6, which was composed of two nearly identical hemispheres. The TNT equivalent is 10 kg. The explosion origin or center is 1200 mm from both the centers of the RC slabs and the pressure tool and 700 mm above the ground. Explosives are positioned by the nylon rope hanging on the lifting bracket and fixed by the mesh pocket and white cloth belt to ensure that the ball center is at the center of the component’s blast surface. In addition, a metal shield is also set behind the component, which is welded by several steel plates to prevent shock wave damage to the test sensors and cables behind the component.

Figure 3 shows the layout of the pressure, acceleration and displacement sensors used in the experiment.

Considering that the distribution of blast loads at different positions of the RC slabs subjected to close-in blast loading differs greatly, four PCB piezoelectric pressure sensors, numbered P1~P4, are arranged according to symmetry. Among them, the measuring point P1 is facing the explosion center, which is taken as the origin of the coordinate system, where the horizontal direction is the X-axis and the vertical direction is the Y-axis. The parameters of the pressure-measuring points are listed in Table 1.

The displacement is measured by the contact pen-type displacement sensors, which are arranged on the back surface of the slabs. The embedded parts are fixed in the holes drilled beforehand using anchoring adhesive. The embedded parts and sensors are in threaded connection. Each slab is equipped with five displacement sensors, which are arranged at intervals along the length of the slab.

### 2.2. Experimental Results and Analysis

#### 2.2.1. Blast-Load Measurement

The blast loading-time history of the measuring points P1~P3 are shown in Figure 4.

It can be seen from the figure that the trend of the reflected pressure waveforms at different measuring points are basically the same, exhibiting the typical law of exponential attenuation. Specifically, as P1 is closest to the explosion center, the reflected pressure here has a typical double-peak structure, and the peak overpressure of the next peak crest is remarkably higher than that of its previous peak crest. The analysis shows that the first wave crest is caused by the reflection of the incident wave on the wall, and the second one may be induced by the action of high-temperature detonation products on the sensitive surface of the sensor. Then the characteristic parameters of each measured curves are extracted as shown in Table 2.

From the table, the peak overpressure and specific impulse of the shock waves decrease with the increasing scaled distance, the peak positive pressure and specific impulse of normal reflection are the largest, and the blast load distributed on the surface of the RC slabs is obviously uneven. Figure 4 gives the overpressure-time history obtained based on the Kingery–Bulmash [25] method to calculate the blast parameters from the air explosion. This empirical model has been embedded into a variety of calculation programs such as ConWep and LS-DYNA (i.e., the keyword *LOAD_BLAST) and has high reliability. In general, the peak overpressure and positive pressure action time of the measured overpressure-time history and the ConWep pressure-time history are relatively consistent, showing good agreement.

#### 2.2.2. Displacement Response-Time History

Figure 5 presents the measured displacement-time history of slab A.

The displacement data represent the displacement of the slab along the propagation direction of the shock wave, and the negative sign represents the displacement of the slab toward the detonation center. The peak displacement and residual displacement of each measured displacement curve of the slab A are extracted, as shown in Table 3.

It can be seen from the table that when RC slab A with the low reinforcement ratio is subjected to explosion shock waves, its peak displacement at the center can reach 1.98 cm and the residual displacement is about 1.73 cm. The maximum displacement of the slab at the position farther away from the center gradually decreases. It can be seen from the diagram that with the increase of the distance from the center of the plate, the peak displacement of the plate gradually decreases, and with structural vibrations, the slab rebounds to some extent after reaching the peak displacement. The rebound is most obvious at D1 near the support, with a negative displacement of about 8 mm. It is important to note that both the distance from D5 and D1 to the support is basically the same, but as the ground has enhancement effect on shock waves, the blast loading strength below the slab is higher than that above the slab, resulting in some difference in the displacement response between the two points.

Similarly, Figure 6 shows the measured displacement curves of slab B with the high reinforcement ratio.

The peak displacement and residual displacement of each measured displacement curve of slab B are extracted, as shown in Table 4.

It can be seen from the table that the peak displacement of the D8 measuring point in the center of the plate is 1.41 cm and the residual displacement is about 0.58 cm. The farther the measuring point is from the center of the plate, the smaller the positive displacement; the larger the rebound displacement is, and the longer the time to reach the maximum positive displacement peak, which conforms to the general deformation law of rectangular RC plate under an explosive shockwave.

The comparison of Figure 5 and Figure 6 shows that the reinforcement ratio of the component will significantly affect the dynamic response of the RC plate under a shockwave. The peak displacement and residual displacement of the slab with high reinforcement ratio are smaller than that of the slab with a low reinforcement ratio, and the overall displacement response rebounds to a large extent, indicating that the degree of damage of the slab with a high reinforcement ratio is relatively small, and the deformation recovery ability is still strong after the shock wave.

#### 2.2.3. Comparison of Damage Modes

Figure 7 gives a comparison of the damaged modes of the two RC slabs after the experiment.

The impact faces of the two plates are kept intact. As can be seen, there are multiple longitudinal and transverse cracks on the back surface of slab A, obvious oblique shear cracks on both ends of the slab, broken concrete in the middle near the edge, and exposed reinforcing bars (rebars), which means serious damage. There are only small longitudinal and transverse cracks appearing on the back surface of slab B, and the crack width is significantly smaller than that of slab A. There is no obvious concrete crushing phenomenon, which is mild damage.

## 3. Numerical Simulation

### 3.1. Finite Element Model

To further study the anti-explosion performance of the RC slabs under close-in blast loading, a finite element model (FEM) is constructed using the multi-material fluid-structure coupling algorithm, as shown in Figure 8.

The model is composed of rebars, concrete, clamps, and air. The separate modeling method is employed for the RC slabs, and the rebars and concrete are coupled through the keyword *CONSTRAINED_BEAM_IN_SOLID [26]. The rebars are modeled with beam element, and the rest with 3D solid element. The explosive and air are modeled using Eulerian grids, and the clamps and concrete slabs using Lagrangian grids. The keyword *RIGIDWALL_PLANAR [26] is adopted to define the influence of using a rigid wall to simulate the ground. The reflection-free boundary condition is used for the air domain boundary, and the model’s unit system is m-kg-s.

### 3.2. Material Model

The explosive is described with the key word *MAT_HIGH_EXPLOSIVE_BURN [27] and the JWL equation of state (EOS), which is expressed as:(1)$$P=A\left(1-\frac{\omega }{R_{1}V}\right)e^{-R_{1}V}+B\left(1-\frac{\omega }{R_{2}V}\right)e^{-R_{2}V}+\frac{\omega E}{V}$$
where *P* is the pressure of detonation products, *V* is the relative volume, *E* is the initial internal energy density, and *A*, *B*, *R*_1_, *R*_2_, and *ω* are the EOS parameters. The JWL EOS parameters of the TNT explosive can be found in reference [25].

The air is defined by the combination of *MAT_NULL and *EOS_LINEAR_POLYNOMIAL equation of state [27], and the linear polynomial EOS is given by:(2)$$P=C_{0}+C_{1}\mu +C_{2}\mu^{2}+C_{3}\mu^{3}+\left(C_{4}+C_{5}\mu +C_{6}\mu^{2}\right)E_{1}$$
where $\mu =\frac{\rho }{\rho_{0}}-1$, ρ represents the current air density, $\rho_{0}$ is the initial air density, *P* is the air pressure, *C*_0_~*C*_6_ are the coefficients of the polynomial equation, *E*_1_ is the internal energy density, and *V*_0_ is the initial relative volume. The parameters for air can be seen in reference [25].

The steel clamps, two-way HRB400 steel bars, and HPB235 stirrups are simulated with a type 3 bilinear elasto-plastic model (*MAT_PLASTIC_KINEMATIC) [27] in LS-DYNA, which is an isotropic-kinematic hardening or mixed isotropic-kinematic hardening model that can approximately simulate the elasto-plastic stage of rebars, and simplify the plastic and hardening stages into an oblique line [17]. The relevant material parameters are listed in Table 5.

The concrete material is defined by the keyword *MAT_CONCRETE_DAMAGE_REL3 [27]. In the model, three strength failure surfaces (initial yield surface, ultimate strength surface, and softening strength surface) are employed to describe the plastic properties of concrete materials, which takes into account the characteristics of such materials such as elastic fracture energy, strain rate effect, and restraint effect. The input only requires three parameters of the concrete material, namely density, uniaxial ultimate compressive strength, and Poisson’s ratio, and the remaining parameters are automatically generated by the system [27]. The material parameters include density $\rho_{0}^{}=$ 2300 g/cm^−3^, uniaxial compressive strength $f_{c}^{\prime }=30$ MPa, and Poisson’s ratio ν=0.2. Under blast loading, the strain rate of the reinforced concrete is up to 100~10,000 s^−1^. The strength of the materials under dynamic loading is fundamentally different from that under quasi-static loading. Therefore, the strain-rate effect of materials needs to be considered for blast-loading analysis. In the material model, the defined strain-rate curve can be called by LCRATE.

### 3.3. Finite Element Model Verification

Spherical explosive blasts propagate outward in the form of spherical shockwaves in free air, so the one-dimensional spherically symmetric model can be used to simulate the three-dimensional diffusion of blast shockwaves, which will greatly reduce the number of grids and improve the computational efficiency. Figure 9 shows the variation law of peak overpressure with scaled distance calculated by models with different grid sizes. From the figure, as the scaled distance increases, the sensitivity of peak overpressure to grid size gradually decreases.

When the scaled distance is less than 0.8 kg/m^3^, the peak overpressure is highly sensitive to grid size and has great discreteness, indicating that the grids need to be fine enough to ensure the computational accuracy. Considering the size of the three-dimensional model, and the computational accuracy and efficiency, the 10 mm grid size is selected for the subsequent finite element modeling and analysis.

To further verify the accuracy of the FEM, the time-history curve of shockwave overpressure obtained by finite element calculation is compared with the measured results, as shown in Figure 10.

It can be seen from the figure that the measured results are in good agreement with the calculated pressure curves. The calculated values and measured values of the peak overpressure at P2 and P3 have a deviation of 0.7% and 5%, respectively, and for the corresponding specific impulse at the two points, the deviation between the calculated and measured values is 27% and 13%, respectively.

Figure 11 gives the comparison between the measured displacement curves and the calculated results of D3 and D4.

According to the figure, the peak displacement arrival time of the calculated curve is slightly earlier than the measured result, and the calculated peak displacement is larger than the measured value, but the residual displacement error of the two values is less than 1%, so the results have good agreement. Considering that the residual displacement of the RC slabs is the permanent displacement of the structure after the blast shockwaves, it can better characterize the degree of plastic damage of the slabs compared to the peak displacement. Therefore, it can be concluded that the calculation results can accurately reflect the dynamic mechanical response of the RC slabs. On the whole, the FEM proposed in this paper can accurately reflect the dynamic response characteristics of the RC slabs and the calculation results are reliable.

### 3.4. Analysis of Finite Element Calculation Results

#### 3.4.1. Explosion Load Analysis

Based on the simulation results, the load distribution on the impact surface of the RC slabs at the scaled distance of 0.56 m/kg^1/3^ is obtained, as shown in Figure 12. In the figure, the X-axis and Y-axis represent the directions of the short span and long span of the slab, respectively.

According to Figure 12a, the peak overpressure on the impact surface of the RC slab is large in the center and gradually decreases at both ends along the long-span direction. Due to the ground reflection effect, the peak overpressure at the end near the ground is slightly larger than that at the top, but along the short-span direction it is basically the same. Different from the peak overpressure distribution, the specific impulse of the impact surface of the RC slab in Figure 12b gradually reduces along the long-span direction, and the impulse near the ground is about two times that at the top. It shows that the ground makes the shockwave appear to be strengthening and creates a convergence effect at the bottom of the component. Therefore, the load distribution in the RC slab in the close-in explosion experiment is affected by many factors, such as the ground, the distance from the explosion center, and the angle of incidence. Although peak overpressure and impulse are unevenly distributed, there exists a distribution law.

#### 3.4.2. Damage Analysis of RC Slab

The contour plots of the plastic damage of the two slabs are presented in Figure 13, where (a) front face and (b) rear face.

As can be seen from the figure, the damage distribution of the two RC slabs is similar. Taking the damage distribution of the blast face as an example, whether it is plate A or plate B, the damage is basically distributed along the reinforcement around the frame and the lateral constraint direction. The damage distribution of RC slabs is mainly along the transverse and longitudinal reinforcement and is obviously affected by frame constraints. The damage degree of RC plates with different reinforcement ratios is significantly different. The damage range and damage degree of the plates with small reinforcement ratios are significantly higher than those of the plates with large reinforcement ratios, regardless of the blast front or back surface. There is a large rectangular plastic strain zone in the center of the back blasting surface of the small reinforcement ratio plate, and several plastic strands appear along the short-span direction. The concrete in some areas is in almost complete failure, showing typical bending failure characteristics.

#### 3.4.3. Residual Displacement Analysis

Figure 14 shows curves of the residual displacement distribution of the two slabs along the long-span direction.

From the figure, the curves of their residual displacement along the long-span direction under shockwave loading take the shape of a parabola, but the displacement of slab A is obviously smaller than that of slab B. At the symmetrical position along the slab center, the displacement at the end near the ground is slightly larger than that at the other end, mainly because of the ground reflection effect, which is consistent with the phenomenon observed in the experiment.

#### 3.4.4. Residual Bearing Capacity Analysis

To further quantitatively evaluate the degradation of the bearing capacity of damaged RC slabs, the residual bearing capacity of damaged RC slabs was simulated by restarting in LS-DYNA, and the quasi-static loading was carried out by slowly applying the displacement perpendicular to the panel at each node of the component face. The method of applying a displacement load is shown in Figure 15.

Extract the reaction force at the support, and finally get the curve of the residual bearing capacity of the damaged member with displacement, as shown in Figure 16.

From Figure 16, the residual load-bearing capacity curves of the two damaged RC slabs under the same blast-loading conditions are obviously different. For slab A, when the mid-span displacement increases to 20 mm, its bearing capacity reaches the maximum, about 2000 kN; when the mid-span displacement increases to 50 mm, its bearing capacity almost decreases to 0, indicating that the slab has been completely damaged at this time. For slab B, when the mid-span displacement increases to 45 mm, its bearing capacity reaches the maximum, around 2250 kN, and when the displacement continues to increase to 105 mm, its bearing capacity is close to zero. The above data fully shows that increasing the reinforcement ratio can not only ensure that the RC slab has a high residual bearing capacity after explosive loading, but also ensures that the damaged members have better ductility and good energy-absorption effect.

Figure 17 gives the comparison of the load-bearing capacity of the two RC slabs before and after the blast-loading test.

It can be seen from the figure that the shape of the bearing capacity-displacement curves of the RC slabs after blast loading have changed to some extent; specifically, the peak bearing capacity decreases to varying degrees, and the mid-span displacement corresponding to the peak bearing capacity increases, which is mainly due to the degradation of the bending stiffness after cracks appear in the damaged slabs. From the values, the bearing capacity of slab A decreases by 750 kN, or 26% compared to the undamaged slab; the residual load-bearing capacity of slab B decreases by 600 kN, or 20% compared to the undamaged slab, which is slightly smaller than that of slab A.

## 4. Conclusions

In this paper, the blast-loading experiment and numerical simulation were performed for reinforced concrete (RC) slabs with two typical reinforcement ratios. The environmental parameters of relevant loads and the damage data of displacement response of the specimens were obtained, and the calculation model was verified based on the measured data. Finally, the residual load-bearing capacity of the damaged RC slabs was analyzed. The main conclusions are as follows:The load distribution in the RC slabs under close-in blast loading was extremely uneven. The peak overpressure on the impact surface of the slabs was large in the center along the long-span direction and gradually decreased at both ends. The specific impulse was gradually reduced along the long-span direction. The impulse near the ground was about two times that at the top.The damage distribution of the two RC slabs with different reinforcement ratios was similar, but the degree of damage differed markedly. A large rectangular plastic strain zone appeared in the center of the back surface of the slab with a low reinforcement ratio, several plastic strands along the short-span direction could be observed, and the concrete in some areas almost completely failed, and the damage range and degree are significantly higher than those of the slab with a high reinforcement ratio.Compared with the undamaged slabs, the shape of the resistance curves of the damaged RC slabs saw significant changes, and their load-bearing capacity and bending stiffness were irreversibly degraded. Increasing the reinforcement ratio can not only inhibit the crack extension and reduce the residual displacement of components, but also reduce the decrease of bearing capacity after damage.

## Figures and Tables

**Figure 1 materials-15-06449-f001:**
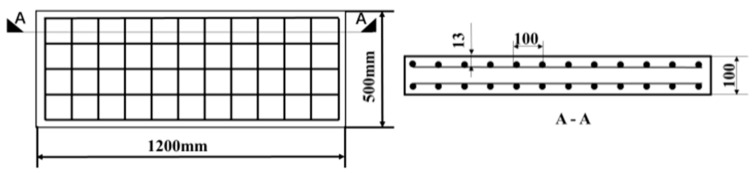
Schematic drawings of RC slab reinforcement.

**Figure 2 materials-15-06449-f002:**
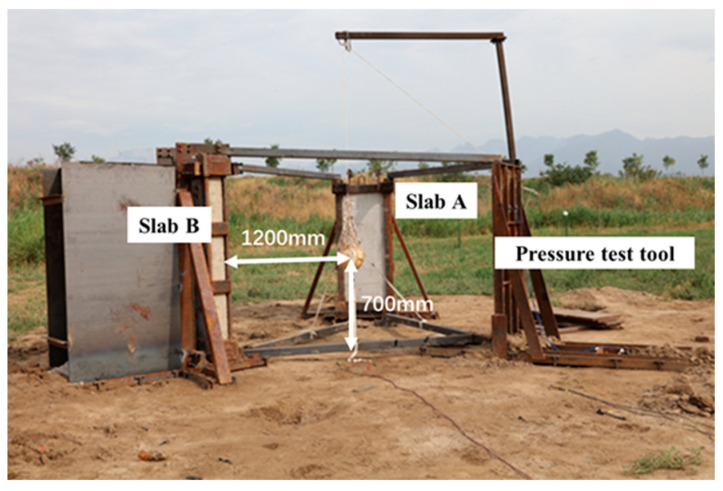
Test site layout.

**Figure 3 materials-15-06449-f003:**
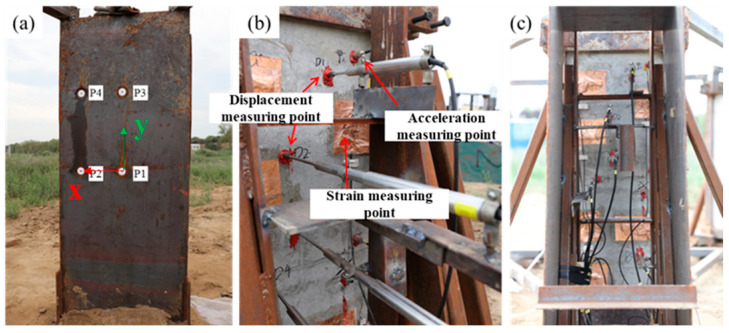
Layout of pressure and displacement measuring points, (**a**) the layout of the pressure sensors, (**b**) the layout of the acceleration and displacement sensors, and (**c**) overall survey point distribution.

**Figure 4 materials-15-06449-f004:**
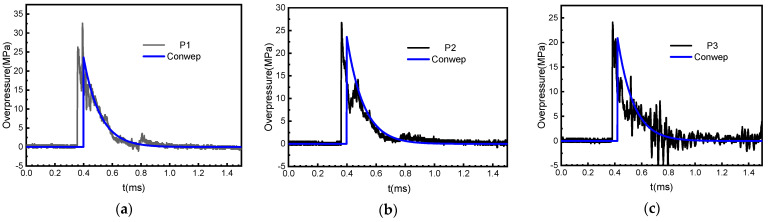
Comparison between measured overpressure curves of shock waves at different measuring points and ConWep calculation results: (**a**) the blast loading−time history of the measuring point P1, (**b**) the blast loading−time history of the measuring point P2, and (**c**) the blast loading−time history of the measuring point P3.

**Figure 5 materials-15-06449-f005:**
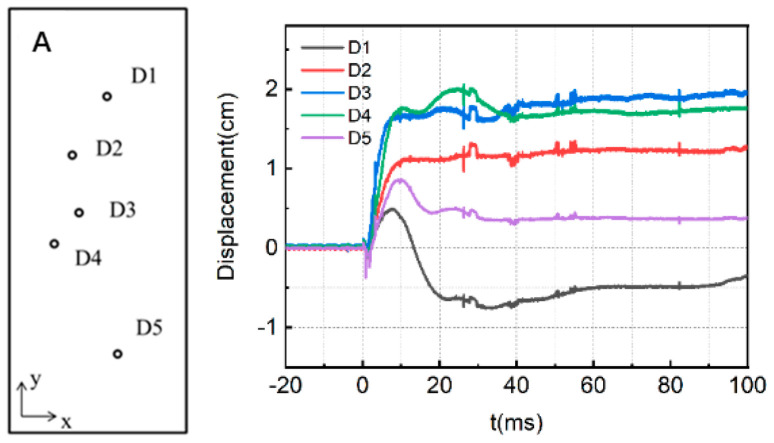
Measured displacement−time history of slab A.

**Figure 6 materials-15-06449-f006:**
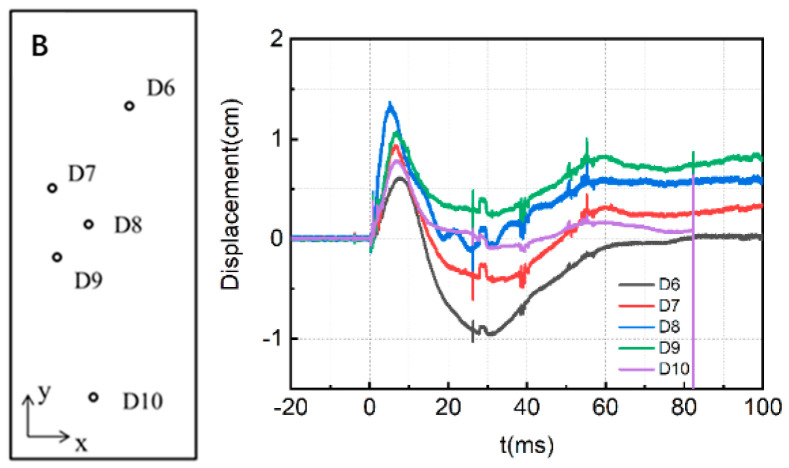
Measured displacement−time history of slab B.

**Figure 7 materials-15-06449-f007:**
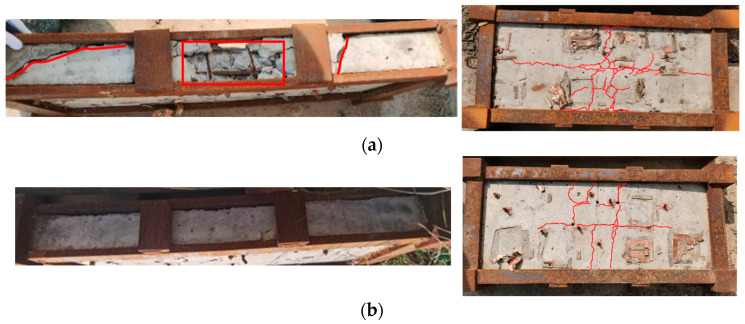
Comparison of damage modes of the two RC slabs: (**a**) damage effect of slab A (low reinforcement ratio), and (**b**) damage effect of slab B (high reinforcement ratio).

**Figure 8 materials-15-06449-f008:**
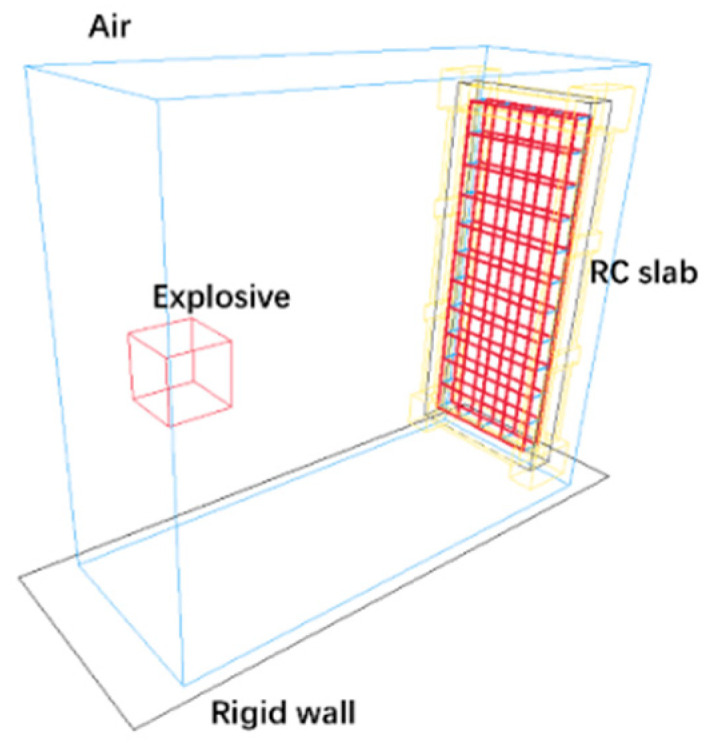
Finite element model.

**Figure 9 materials-15-06449-f009:**
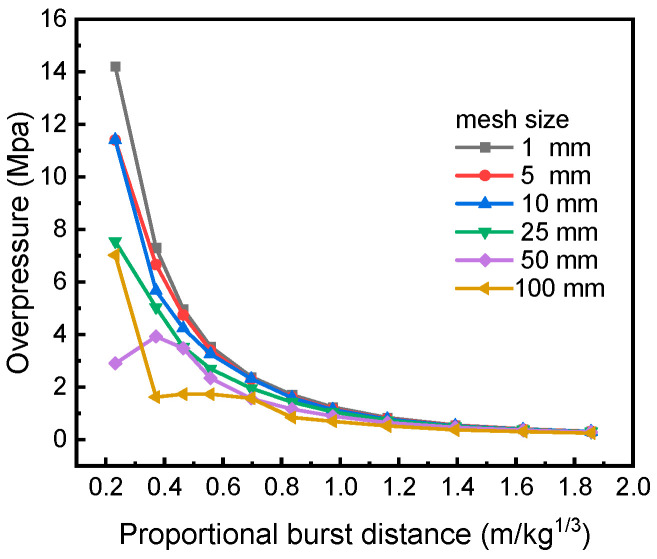
Comparison of calculation results under different grid sizes.

**Figure 10 materials-15-06449-f010:**
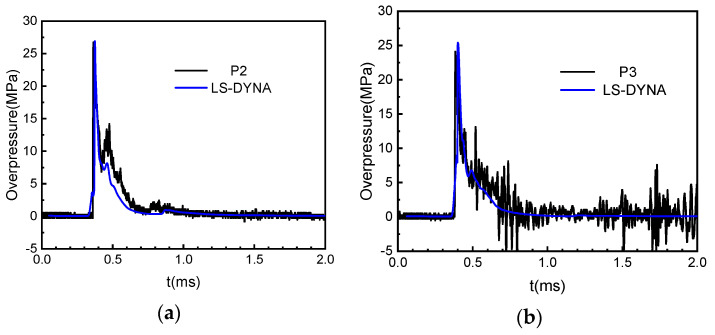
Comparison of pressure−time history, (**a**) the comparison of pressure−time history of the measuring point P2, (**b**) The comparison of pressure−time history of the measuring point P3.

**Figure 11 materials-15-06449-f011:**
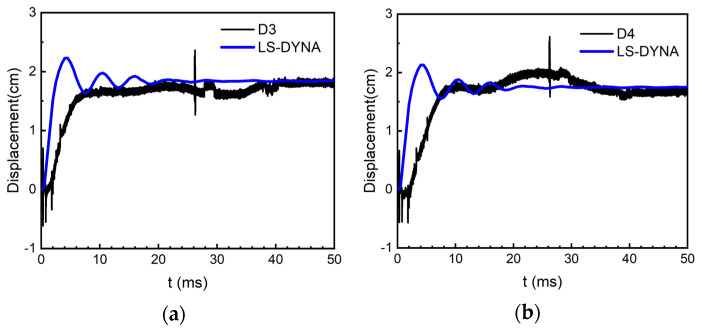
Comparison of displacement−time history: (**a**) the comparison of displacement−time history of the measuring point D3, and (**b**) the comparison of displacement−time history of the measuring point D4.

**Figure 12 materials-15-06449-f012:**
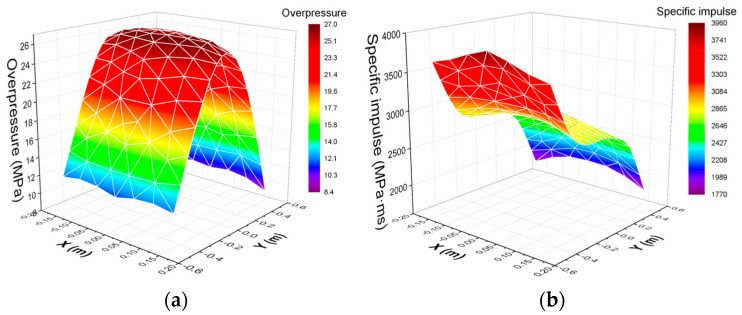
Contour plots of load distribution in RC slab: (**a**) peak overpressure, (**b**) specific impulse.

**Figure 13 materials-15-06449-f013:**
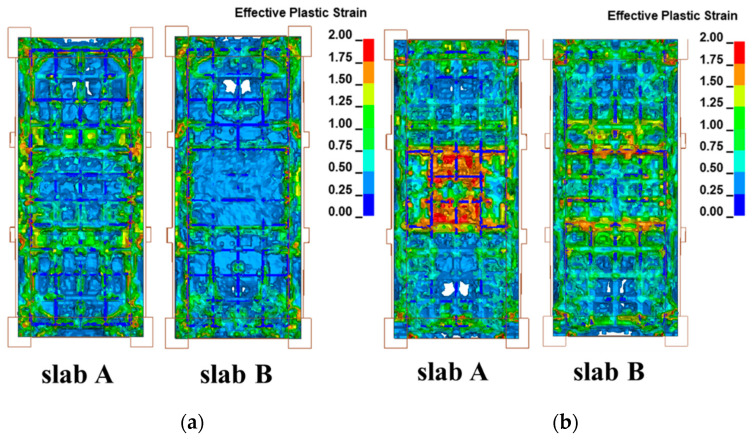
Comparison of damage contours of slab A and slab B: (**a**) Front face, (**b**) Rear face.

**Figure 14 materials-15-06449-f014:**
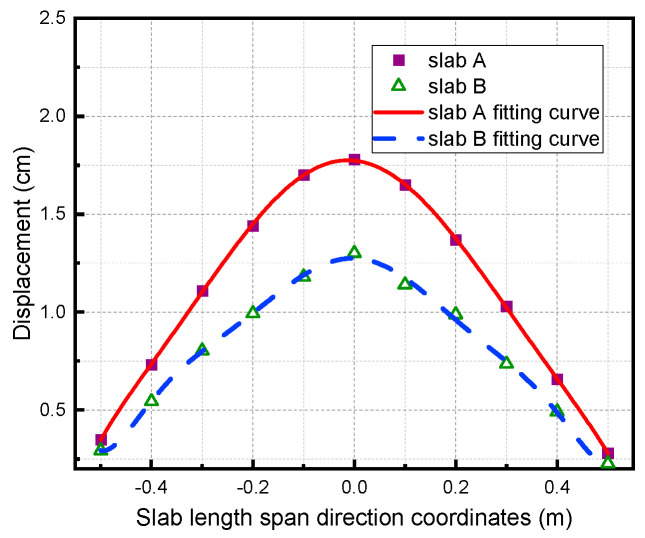
Comparison of residual displacement of the RC slabs along the long−span direction.

**Figure 15 materials-15-06449-f015:**
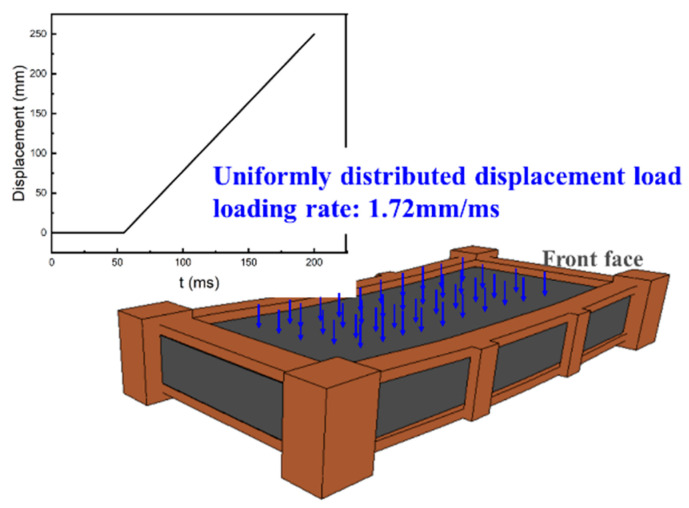
Schematic diagram of application method of displacement load.

**Figure 16 materials-15-06449-f016:**
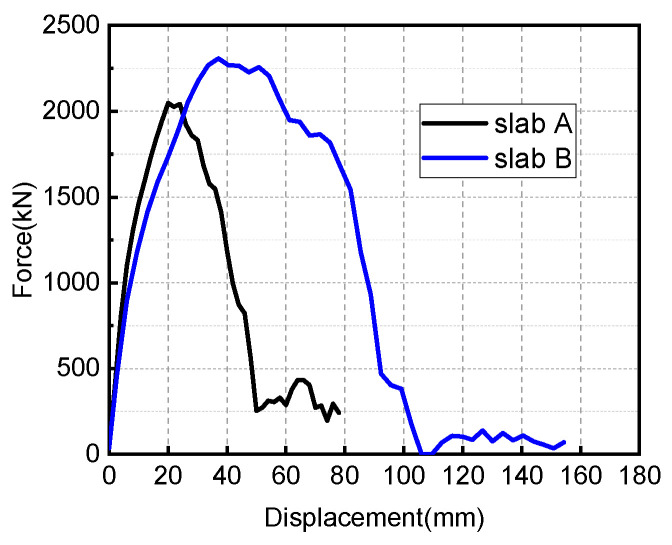
Residual load-bearing capacity curves of the two damaged RC slabs.

**Figure 17 materials-15-06449-f017:**
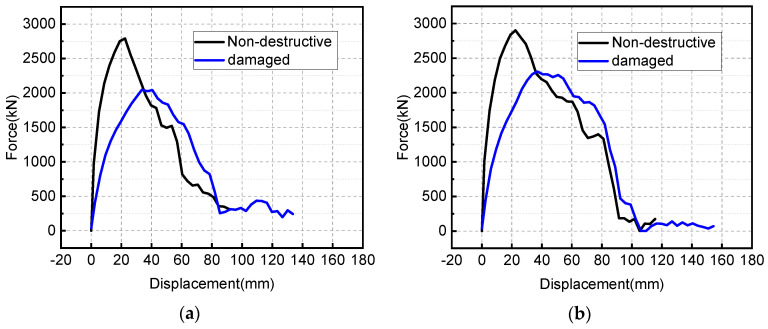
Comparison of load−bearing capacity between undamaged and damaged slabs: (**a**) slab A with low reinforcement ratio, and (**b**) slab B with high reinforcement ratio.

**Table 1 materials-15-06449-t001:** Coordinates of the pressure-measuring points.

Measuring Point	X/m	Y/m	Distance from Explosion Source/m	Scaled Distance/m/kg^1/3^	Angle of Incidence/°
P1	0	0	1.2	0.557	90
P2	0.16	0	1.211	0.562	82.4
P3	0	0.3	1.237	0.574	75.96
P4	0.16	0.3	1.247	0.579	74.18

**Table 2 materials-15-06449-t002:** Characteristic parameters of pressure measuring points.

Measuring Point	Scaled Distance/m/kg^1/3^	Peak Positive Pressure/MPa	Shock Wave Arrival Time/ms	Specific Impulse/MPa × ms	Angle of Incidence/°
P1	0.557	32.32	0.36	3.35	90
P2	0.562	26.47	0.363	3.02	82.4
P3	0.574	23.58	0.38	2.93	75.96

**Table 3 materials-15-06449-t003:** Parameters of slab A displacement measuring points.

Displacement Measuring Point	D1	D2	D3	D4	D5
Peak displacement/cm	0.51	1.18	1.97	1.98	0.89
Residual displacement/cm	−0.37	1.18	1.97	1.73	0.38

**Table 4 materials-15-06449-t004:** Parameters of slab B displacement measuring points.

Displacement Measuring Point	D6	D7	D8	D9	D10
Peak displacement/cm	0.67	0.92	1.41	1.09	0.79
Residual displacement/cm	0.03	0.31	0.58	0.80	0.08

**Table 5 materials-15-06449-t005:** Material parameters of rebars and clamps.

Parameter	*ρ*/kg·m^−3^	*E*/GPa	*V_s_*	*σ_y_*/MPa	*E_t_*/GPa	*C*/s^−1^	*P_s_*	*F_s_*
HRB400	7850	210	0.28	400	2.1	40	5	0.2
HPB235				235				
Clamp				300		0	0	

## Data Availability

The study did not report any data.

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
