# Peer review of "Blast Resistance of Reinforced Concrete Slabs Based on Residual Load-Bearing Capacity"

_materials, 2022, doi:10.3390/ma15186449_

Round 1

Reviewer 1 Report

 The paper “Blast Resistance of Reinforced Concrete Slabs Based on Residual Load-Bearing Capacity” presents an interesting study, but the writing and presentation are very poor. Please see some of my specific comments.

1.   The first 4 lines in the abstract are unnecessary, either remove them or rewrite them briefly.

2.       Line 29, revise the statement, The reinforced concrete (RC) slab is a typical and key load-bearing structural member in reinforced concrete buildings. Typical and key are conflicting here. Also, the introduction is poorly written and did not include much present research.

3.    Please mention the code used to design the slab in the Experimental study section. A slab does not require any stirrup, stirrup is applicable for a beam to resist shear force, please explain.

4.       Methodology section should be written to make it clearer.

5.       Replace the photos in Figure 7 With high-resolution photos.

6.    How did you select low and high reinforcement ratios for the RC slabs, please mention them in the paper.

7.     The whole paper is poorly written, and it did not demonstrate the cause and effects of the outcome of the study.

8.     In conclusion, at point 2, don’t write slabs A and B, rather specify the reason for the outcome. Also, mention what makes the difference. 

Reviewer 2 Report

The paper is well written and is of importance to the readers. It may be accepted for publication. However, the authors are advised to improve the Introduction part of the article and add some more literature. Also, explain why two typical reinforcement ratios have been chosen for the study. 

Reviewer 3 Report

The authors made blast loading experiments and numerical simulations for reinforced concrete (RC) slabs with two typical reinforcement ratios. They obtained by measuring the relevant loads of the damage data of the RC and verified the proposed model based on the measured data, analyzing the residual load-bearing capacity of the damaged RC 

The main conclusion of this study was the following: the damage distribution of the two RC slabs with different reinforcement ratios was similar, but the degree of damage differed markedly. 

Increasing the reinforcement ratio can inhibit the crack extensions and reduce the residual displacement of components and most importantly, can reduce the decrease of bearing capacity after damage. In my opinion, this study is important for designing industrial buildings that must resist a possible undesired explosion of a component from a factory and also for design of the resistance structure of the human shelters that must withstand bombings, in case of war. For this reason, I recommend this article for publishing. However, in the manuscript  must be made minor corrections  before publishing as follow:

1)in the abstract the abbreviations must be avoided;

2) Subchapter 2.2. must be moved on a new line;

3) at row 57: authors must write in the bracket what represents  P-I;

4) Sentence at the rows 192-195 must be revised;

5) The dimensions of greek fonts used for parameters writing from the rows (249-250 ) must be reduced;

6) The references from the end of the article must be written according to the MDPI style.

Author Response

请参阅附件

Round 2

Reviewer 1 Report

Authors have addressed my comment. I have no further technical comments but the Englsh of the paper needs to be improved.